# Visualizing Intramolecular Dynamics of Membrane Proteins

**DOI:** 10.3390/ijms232314539

**Published:** 2022-11-22

**Authors:** Tatsunari Ohkubo, Takaaki Shiina, Kayoko Kawaguchi, Daisuke Sasaki, Rena Inamasu, Yue Yang, Zhuoqi Li, Keizaburo Taninaka, Masaki Sakaguchi, Shoko Fujimura, Hiroshi Sekiguchi, Masahiro Kuramochi, Tatsuya Arai, Sakae Tsuda, Yuji C. Sasaki, Kazuhiro Mio

**Affiliations:** 1AIST-UTokyo Advanced Operando-Measurement Technology Open Innovation Laboratory (OPERANDO-OIL), National Institute of Advanced Industrial Science and Technology (AIST), 6-2-3 Kashiwanoha, Chiba 277-0882, Japan; 2Graduate School of Medical Life Science, Yokohama City University, 1-7-29 Suehiro-cho, Tsurumi-ku, Yokohama 230-0045, Japan; 3Graduate School of Frontier Sciences, The University of Tokyo, 5-1-5 Kashiwanoha, Chiba 277-8561, Japan; 4Center for Synchrotron Radiation Research, Japan Synchrotron Radiation Research Institute, 1-1-1 Kouto, Sayo-cho, Hyogo 679-5198, Japan; 5Graduate School of Science and Engineering, Ibaraki University, Hitachi 316-8511, Japan

**Keywords:** single-molecule analysis, conformation dynamics, membrane proteins, diffracted X-ray tracking technique

## Abstract

Membrane proteins play important roles in biological functions, with accompanying allosteric structure changes. Understanding intramolecular dynamics helps elucidate catalytic mechanisms and develop new drugs. In contrast to the various technologies for structural analysis, methods for analyzing intramolecular dynamics are limited. Single-molecule measurements using optical microscopy have been widely used for kinetic analysis. Recently, improvements in detectors and image analysis technology have made it possible to use single-molecule determination methods using X-rays and electron beams, such as diffracted X-ray tracking (DXT), X-ray free electron laser (XFEL) imaging, and cryo-electron microscopy (cryo-EM). High-speed atomic force microscopy (HS-AFM) is a scanning probe microscope that can capture the structural dynamics of biomolecules in real time at the single-molecule level. Time-resolved techniques also facilitate an understanding of real-time intramolecular processes during chemical reactions. In this review, recent advances in membrane protein dynamics visualization techniques were presented.

## 1. Introduction

Membrane proteins are expressed in cell and organelle membranes and play important roles in various physiological functions, such as ion/chemical transport, signal transduction across the membrane, and mechanosensing including heat and force. To understand protein functions, it is necessary to understand protein structure and dynamics. X-ray crystallography, nuclear magnetic resonance (NMR) spectroscopy, and cryo-electron microscopy (cryo-EM) techniques have been mainly used to analyze membrane protein structure. X-ray crystallography uses X-ray diffraction. Due to its high resolution and well-established workflows, X-ray crystallography was widely used for structure determination until the recent revolution in cryo-EM. NMR uses strong local magnetic fields to analyze the arrangement of nuclei in atoms. NMR applications depend on the size of molecules due to their complexity, but it can detect structure and dynamics in solution [1]. Cryo-EM single-particle analysis (SPA) has emerged as one of the most effective techniques in structural biology [2]. This “resolution revolution” was achieved by developing direct detection device (DDD) cameras, which can detect electrons directly at higher frame rates than charge-coupled device (CCD) cameras. DDD recording has high detective quantum efficiency. It enables movie recording during exposure and the correction of beam-induced image movement.

In contrast, the available techniques for the direct observation of intramolecular motion are limited. Because the conformational dynamics of membrane proteins occur on nanosecond to millisecond timescales, high-speed recording is required. If the target protein has multiple conformations, the gross dynamics observed is a mixture of several motion components with various timescales, and structural determination can also be ambiguous. Similarly, the simultaneous determination of protein structure and molecular dynamics is extremely difficult.

Single-molecule measurement using an optical microscope has been widely used for kinetic analysis, where a fluorescent probe is attached to a target protein, and the movement of the fluorescent signal is observed as a corresponding motion of the target molecule. The development of high-quality detection cameras provides sufficient spatial resolution to monitor molecular dynamics. However, the wavelength of light limits this system’s spatial resolution to a few hundred nanometers.

Optical tweezers (OT) use light pressure to trap or move single molecules and have been applied to single-molecule manipulation with ~0.2 nm spatial resolution and ~10 μs temporal resolution [3,4,5]. Because OT manipulate proteins in the light pass, OT can help us understand folding and unfolding mechanisms [6], protein-membrane interactions [7], and the ligand-induced conformational fluctuations of proteins [8].

X-rays and electron beams have been used to obtain higher-order spatiotemporal information about protein dynamics. Recent advances in image processing and X-ray/electron beam technology have resulted in the development of various dynamic determination methods. X-ray free electron laser (XFEL) and cryo-EM SPA have recently been used for structural analysis. These techniques can reveal protein structure at the atomic level and provide dynamics information by rearranging structural information from snapshots. Combined with the pump-and-probe method, XFEL provides structural changes at the atomic level while avoiding radiation damage. Diffracted X-ray tracking (DXT) can elucidate molecular dynamics from tilting (*θ*) and twisting (*χ*) motion [9,10]. In DXT, specific domains of a single protein are labeled with gold nanocrystals, and the movement of X-ray diffraction spots generated from gold nanocrystals reflects the dynamics of the labeled domains.

Another approach for dynamics recording is high-speed atomic force microscopy (HS-AFM). Atomic force microscopy (AFM) was developed in 1986 to scan the surfaces of insulators on an atomic scale [11]. However, it is difficult to observe the behavior of biomolecules in real-time using conventional AFM as it takes more than 30 s to capture an image. HS-AFM is a scanning probe microscope that can capture the structural dynamics of biomolecules in real time at the single-molecule level. HS-AFM provides nanometer structural information and dynamics in biological samples at subsecond resolution. Recently, a sophisticated AFM technique using background threshold subtraction has made it possible to obtain accurate volume information in the oxidation process of ferredoxin-dependent flavinthioredoxin reductase [12]. The correlation of AFM measurements with super-resolution fluorescence microscopy to achieve excellent lateral resolution practically at the structured illumination diffraction limit was also recently reported [13]. These emerging techniques for AFM may be implemented in HS-AFM in the near future.

This review presents the history and recent advances in techniques for visualizing the intramolecular dynamics of membrane proteins. In contrast to well-established structural analysis techniques, the techniques for monitoring protein dynamics in real time are limited. This is an emerging research area and can be expanded in the future. We introduce the dynamics analysis methods for membrane proteins that have been rapidly developing in recent years.

## 2. Measurement of Observing Dynamics of Membrane Protein

### 2.1. Single Molecular Dynamics Using Optical Measurement

Single-molecule techniques have been extensively used to analyze structural rearrangements in the reaction cycle of motor proteins that involve energy-consuming processes. F_1_-ATPase is a biological rotary motor protein that couples transmembrane proton transport driven by the proton motive force to the synthesis of ATP from adenosine diphosphate and inorganic phosphate. F_1_-ATPase consists of two parts: a membrane-embedded F_0_ motor and a protruding F_1_ stator ring with alternating α and β subunits. In 1997, F_1_-ATPase rotational motion was observed under epifluorescence microscopy by tracking fluorescent actin filaments attached to the γ-subunit of the α_3_β_3_γ subcomplex (Figure 1A) [14,15]. Actin filaments rotated only counterclockwise when viewed from the membrane side. In addition, the γ-attached actin rotated in 120° steps at low ATP concentrations, supporting Boyer’s binding change mechanism model [16].

Single-molecule techniques using visible light can be used to observe large-scale movement. One problem with directly detecting dynamics using fluorescent materials is the resolution limit due to the wavelength. The resolution of hundreds of nanometers is occasionally insufficient to track small movements within single molecules.

An optical microscope uses visible light and a lens system to magnify images of small samples. Any object less than half the wavelength of the microscope’s illumination source is invisible under that microscope. The visible light wavelength range is 400 nm to 700 nm, and an optical microscope’s resolution is limited to about 150 nm, which is not adequate to observe a protein image in detail.

In the case of super-resolution, the general expression showing how to localize the fluorophore is expressed as follows [18]:$$\sigma_{\mu_{i}}=\sqrt{\left(\frac{s_{i}^{2}}{N}+\frac{a^{2}/12}{N}+\frac{8\pi s_{i}^{4}b^{2}}{a^{2}N^{2}}\right)}$$
where *σ_μ_* is the standard error of the mean, *N* is the number of collected photons, *a* is the pixel size of the imaging detector, *b* is the standard deviation of the background, *S_i_* is the standard deviation, and *i* is the x or y direction. However, to observe finer molecular motions, electron beams and X-rays are indispensable.

OT use light pressure to trap or force single molecules and have been applied to single-molecule manipulation (Figure 1B). This technique was developed by Ashkin et al. in 1986 [3] and achieved ~0.2 nm spatial resolution and ~10 μs temporal resolution [4,5]. The OT can trap a single particle by light pressure generated from high convergence beams and measure the force on the particle, which has been applied for measuring protein dynamics. Mechanical single-particle measurement by OT has mainly helped understand the folding and unfolding mechanisms of individual *Escherichia coli* ribonuclease H molecules [6], membrane interactions of C2 domains in synaptotagmin-1 and extended synaptotagmin-2 [7], and ligand-induced conformational fluctuations of diadenosine pentaphosphate [8]. The application of OT to the analysis of membrane proteins has been limited so far, but oscillatory OT, which can apply piconewton forces perpendicularly to the cell membrane, demonstrated that a force in the range of 13–50 pN by plasma membrane indentation causes the influx of extracellular calcium through transmembrane ion channels [19]. OT are a powerful single-molecule tool for measuring protein dynamics at the single-molecule level and can provide dynamic information in micro-to-milli-second resolution. Their resolution is frequently limited by the sensitivity of the force detectors.

### 2.2. Single Particle Analysis Using Cryo-Electron Microscopy

Cryo-EM SPA uses an electron beam of 0.002 nm wavelength at a 300 keV accelerating voltage, which is much shorter than that of visible light. Recent technologies have also enabled the improvement of the signal-to-noise ratio in obtaining data. A DDD camera using complementary metal oxide semiconductor (CMOS) technology successfully suppressed image blurring that occurs when electron beams are converted into photons [20]. The DDD camera captures particle images as movies and corrects beam-induced movement [21,22]. Another key improvement in cryo-EM is the use of Bayesian estimation analysis with conditional probabilities. This improved the efficiency of 3D reconstruction and enabled high-resolution structural analysis using cryo-EM [23].

SPA using cryo-EM reconstructs the protein structure by recording protein images embedded in thin ice and calculating a 3D structure from the obtained 2D images using the Fourier transform [24,25,26,27]. Membrane proteins are usually difficult to crystallize. However, the structures of many membrane proteins have been elucidated using this technique because SPA does not require protein crystallization. These technological developments have determined the structure of the TRPV1 channel at 3.4 Å resolution [28,29]. This was really surprising because TRPV1 is very large (~400 kDa) and difficult to crystalize, as shown by several attempts performed worldwide.

SPA can simultaneously present multiple intermediate structures using image classification techniques. Using a 3D classification technique, Kishikawa et al. obtained multiple conformations of the *Thermus thermophilus* V-ATPase [17,30]. The rotation of the V-ATPase is driven by the free energy difference in the ATP hydrolysis reaction between the two catalytic sites. The obtained intermediate structures revealed dynamic motions by linking each structure (Figure 1C). Three-dimensional classification is also used to discover the structural heterogeneity of proteins in SPA. In Gram-negative bacteria, the β barrel assembly machinery (BAM) is used to fold outer membrane proteins into the membranes. Cryo-EM 3D classification demonstrated large conformational changes in the periplasmic BamB subunit of BAM and the hybridization of EspP β barrel to BamA using curling movement inward to form a barrel-like structure [31].

High-speed sample mixing is another possible method for obtaining structural intermediates using cryo-EM. In 1995, Unwin et al. applied the time-resolved spray method to the cryo-EM analysis of two-dimensional crystals and obtained the open form of the acetylcholine receptor [32]. This technique was applied to SPA, and the millisecond-scale conformational changes were obtained in the calcium-gated potassium channel MthK [33].

Combining molecular dynamics (MD) simulations with SPA can also reveal protein structures complemented by dynamics information. In SPA, conformational information is obtained as snapshots from particle images in different orientations. Dynamics information is affected by structural fluctuations; structures with local fluctuations are averaged and displayed at lower resolution. Matsumoto et al. developed a DEFMap program to extract dynamic information from the electron density map using deep learning and an MD-CNN program to predict dynamics by reconstructing the map [34]. These programs can reduce the cost of dynamics prediction since conventional MD simulation of three test proteins (~200 kDa) took 10–20 h each, while the simulation using DEFMap could be completed in a few minutes. With the recent development in computational technology, calculations using artificial intelligence (AI) and machine learning have become more efficient year by year. By increasing the amount of training data, the functioned analysis of membrane proteins will progress. This, in turn, will accelerate the research and development of new drugs.

### 2.3. Visualizing Protein Dynamics Using High-Speed Atomic Force Microscopy

HS-AFM images the structure of the sample surface by scanning with a sharp tip attached to the free end of a microcantilever and directly observing the dynamics and conformational information of the target protein simultaneously. Unlike steady-state conformations obtained from X-ray crystallography or cryo-EM, HS-AFM can observe the dynamics of raw sample images in an aqueous solution.

Ando et al. increased the scan speed of the AFM for imaging proteins, and the HS-AFM technology was established (Figure 1D) [35,36]. Direct observation of the F_1_-ATPase stator ring revealed the rotating movement of the three β-subunits continuously despite the lack of a γ-subunit [37]. The proper design of all HS-AFM components like scan heads or optimized ultra-short cantilevers is crucial to achieving excellent lateral and temporal resolution [38]. The prototyping and microfabrication of customized cantilevers are required to achieve fast data acquisition times (*t*), which is expressed as follows [39]:$$\;t\geq \frac{2N^{2}}{f_{c}}$$
where *f_c_* is the cantilever resonant frequency and *N* is the number of pixels per line.

Different probe forces can be selected for HS-AFM sample scanning. PIEZO1 is a mechanosensor activated via mechanical stimuli, and its steric structure suggests that the arm domain of PIEZO1 senses the gating force. HS-AFM demonstrated that the pore diameter changed with applied force [40], suggesting a force-dependent conformational flexibility of the pore.

Matin et al. developed high-speed atomic force microscopy line-scanning (HS-AFM-LS) [41]. In contrast to HS-AFM, which scans protein surfaces in two dimensions with a temporal resolution of milliseconds to seconds, HS-AFM-LS scans protein surfaces in only one dimension with a temporal resolution of a few milliseconds. Heath et al. recently developed localization AFM (LAFM) to improve lateral resolution [42]. In the LAFM technique, height values are defined from a local maximum position extracted from the raw data. The values are changed into the probability density per image frame at an atomic level. LAFM observed the pH-dependent conformational changes of CLC-ec1 Cl^−^/H^+^-antiporter in the Angstrom range.

HS-AFM has also been applied to living organisms. Yamashita et al. successfully observed dynamic molecular architectures on the surface of living cells [43], and Fukuma et al. developed 3D-AFM and imaged cell surfaces and intracellular structures in three dimensions using a long needle-like nanoprobe [44]. Combined measurements of HS-AFM with an optical microscope or 3D-AFM were also undertaken.

### 2.4. Time-Resolved Serial Femtosecond Crystallography Using X-ray Free Electron Laser

XFEL was developed as a new commercial X-ray light source in this century. In contrast to conventional synchrotron X-ray light sources, XFEL uses high-intensity and coherently amplified X-ray photons. In conventional X-ray crystal structure analysis, protein crystals are exposed to X-rays for several hundred milliseconds, inevitably causing structural damage to the crystal sample. XFEL uses subpicosecond- to picosecond-order pulsed X-rays, which are shorter than atomic movement after irradiation. Structural information can be obtained by XFEL before irradiation destroys the structure [45,46].

A combination of time-resolved serial femtosecond crystallography (TR-SFX) and the pump–probe techniques are widely used for XFEL dynamics analysis (Figure 1E). In TR-SFX, microcrystals are continuously fed into the XFEL-irradiated area using injectors [47], and the conformational changes of proteins within nanoseconds to milliseconds after laser irradiation can be observed as X-ray diffraction images. Time-resolved conformational changes can be captured by observing samples with different reaction times before X-ray irradiation. The photoexcitation kinetics of bacteriorhodopsin in nanoseconds to milliseconds were captured at the atomic level using TR-SFX [48]. The retinal isomerization-induced intramolecular dynamics and the involvement of water molecules even 10 µs after photoactivation were observed for the bacteriorhodopsin from its snapshots.

Pump-probe techniques can be used in time-resolved experiments using XFEL, especially for light-responsive proteins, including rhodopsin and a photosynthetic reaction center. In ligand-induced conformational dynamics, the mixing rate between protein crystals and a substrate determines their accuracy [49]. Developing a faster sample mixing system will enable time-resolved dynamics measurement of nonphotoresponsive membrane proteins, such as GPCRs, and ion channels.

### 2.5. Diffracted X-ray Tracking

DXT is a single-molecule technique using white X-rays, which can reveal the intramolecular dynamics of target proteins with a two-dimensional axis by tilting (*θ*) and twisting (*χ*) in an aqueous solution [9,10]. In DXT, the specific domain of the target proteins is labeled with gold nanocrystals, and the motions of the proteins are detected as the trajectories generated from the gold nanocrystals (Figure 2A). Intramolecular dynamics have been revealed using DXT, such as for DNA [10], a motor protein [50], and membrane proteins [51,52,53].

X-ray diffractions from gold nanocrystals are generated according to Bragg’s law as follows:2dsinθ=nλ
where *n* is the diffraction order, *λ* is the wavelength of the incident X-rays beam, *d* is the lattice plane spacing for a particular diffracting plane of atoms, and *θ* is the angle between the incident beam and the diffracting plane. DXT requires white X-rays with continuous wavelengths (energy widths ranging from 14.0 to 16.5 keV). Based on Bragg’s law, the angular displacement changes of the diffraction spots can be trackable (Figure 3A,B).

There are three main advantages to DXT. First, DXT can track intramolecular movements with excellent spatiotemporal resolution. The spatial resolution of single-molecule measurements is expressed as *λ*/100, where *λ* is the wavelength. Because the X-rays wavelength used in DXT is 0.01 nm to 0.1 nm, positioning accuracy can be achieved at the picometer level, which is sufficient to recognize the intramolecular movements. In addition, DXT can measure dynamics with microsecond to millisecond order due to the high-brightness X-rays.

Second, DXT can obtain the dynamics data separately via the tilting (*θ*) and twisting (*χ*) motions. The motion of diffraction spots can be continuously tracked, and the detection limit is up to 1 mrad in the *θ* direction and 6 mrad in the *χ* direction. The motion of diffraction spots is analyzed using mean square displacement (MSD), the 2D histogram, and the mean plot of the angular displacements.

Third, DXT can be performed even under solution conditions. Target proteins are immobilized on a polyimide film substrate using chemically anchoring, antigen–antibody reactions, biotin–avidin reactions, or glycoprotein–lectin reactions. Then, gold nanocrystals are attached to the target domain of proteins using the Au–S reaction, the antigen–antibody reaction, etc. Samples in aqueous solution are sandwiched again with polyimide film to perform protein functions at maximum conditions. Here we present examples of dynamics analysis of the TRPV1 channel [53] and 5-HT_2A_ serotonin receptor (5-HT_2A_R) [54,55]. In the case of DXT at the SPring-8 BL40XU beamline, diffraction images were recorded using an X-ray image intensifier (V5445P and V7739P, Hamamatsu Photonics, respectively) and a CMOS camera (FASTCAM SA1.1, Photron, Japan and Phantom V2511, Vision Research, respectively). Gold nanocrystals were deposited on the substrate surface of NaCl (100) or KCl (100) and grown epitaxially to a 40–80 nm diameter to obtain sufficient diffraction signals. The size of the gold nanocrystals is larger than that of proteins, and protein movement is affected by the attached gold nanocrystals in a size-dependent manner [52]. The obtained diffraction spots were tracked by the TrackPy program (v0.3.2 https://doi.org/10.5281/zenodo.60550) (accessed on 17 November 2022) after correcting the background. The trajectories were analyzed using IGOR Pro software (Wavemetrics, Lake Oswego).

TRPV1 is a homotetrameric ion channel that functions as a multiple sensor for heat, capsaicin, and protons [28]. The TRPV1 has a large molecular size of 400 kDa and a complex structure containing more than 20 transmembrane segments. Cryo-EM suggested, in comparing with the apo (PDB:3J5P) and the RTX/DkTx bound open structure (PDB:3J5Q), the extracellular domain of TRPV1 twists to a clockwise direction during channel opening. To clarify the intramolecular dynamics of TRPV1 in channel opening, the extracellular domain (S1–S2 loop) of TRPV1 was labeled with gold nanocrystals, and the molecular rotation was observed at 100 μs per frame (Figure 3C) [53]. A lifetime filtering technique (LT) was applied to segregate the mixed time-scale dynamics. The DXT observed the rotational bias at the extracellular domain from close to open conformational change by capsaicin at 2.5 ≤ LT < 4 ms. Conversely, opposite directional rotational bias was observed by applying the antagonist. The maximum rotation angle of capsaicin-induced TRPV1 was 1.55° at 1 ms intervals to the CW direction. The DXT method directly measured the intramolecular dynamics of the TRPV1 channel for the first time, providing an understanding of the channel opening mechanism.

DXT can be applied to the internal dynamics of membrane proteins in living cells [54,55,56]. 5-HT_2A_R is a Gq-coupled GPCR, which activates phospholipase C. 5-HT_2A_R was expressed in HEK293 cells. Gold nanocrystals were bound at the N-terminus of the extracellular domain via an antibody against the FLAG sequence. In this experiment, the diffracted X-ray blinking (DXB) method using monochromatic X-rays was also used (Figure 2B) [57]. DXT and DXB images were captured at a rate of 100 μs per frame and 100 ms per frame, respectively. X-ray damage to the live cells was eliminated by narrowing the X-ray bandwidth. Depending on the selected recording speed (selection of frame rate), the fast-moving component due to the fluctuation of the receptor itself and the slow-moving component containing the cell signaling cascades were obtained separately. DXT is a very powerful method for clarifying membrane protein dynamics with a high spatiotemporal resolution, even for the intramolecular dynamics in living cells.

The dynamics data obtained from DXT are restricted to movements in labeled domains of the proteins. Therefore, DXT techniques complement other two-dimensional or three-dimensional imaging techniques, such as cryo-EM, HS-AFM, and MD simulation. Combined analysis of these techniques provides the global dynamics of target protein molecules. In addition, DXT can choose suitable frame rates from microseconds to seconds. This means that the directly observed fast molecular motions can be complemented with atomic-level calculations from MD simulation analysis.

The methods for visualizing membrane protein dynamics discussed in this review are summarized in Table 1.

## 3. Discussion and Future Perspectives

Membrane protein dynamics analysis was developed as a single-molecule measurement technique using an optical microscope. Currently, various methods, including cryo-EM, HS-AFM, XFEL, and DXT are now widely used. In HS-AFM, ultrafast unbinding processes of transient biomolecular complexes like streptavidin-biotin were revealed [58]. The nanomechanical characterization revealed by HS-AFM can clarify binding processes and membrane protein functions [59]. Recent advances in image processing technology, the faster sample-mixing system, the time-resolved measurement method, and sufficient knowledge of sample labeling techniques, have made it possible to perform dynamics analysis even with conventional structural analysis methods. Time-resolved measurements reveal the structures of hidden intermediates during membrane protein activation, leading to an understanding of the activation mechanisms of membrane proteins.

Another big trend in structural biology is the use of AI for structural prediction and function assessment. AlphaFold2 [60] and RoseTTAFold [61] are the most well-known structure prediction applications using machine learning. They can predict the structures and biological functions of proteins from amino acid sequences. Recently, structure prediction accuracy has improved significantly, and they are becoming promising tools for elucidating novel mechanisms in proteins of unknown structures. It will not be long before we can freely use information on structure, dynamics, and simulation studies simultaneously to develop new drugs and deepen our understanding of biological functions.

## Figures and Tables

**Figure 1 ijms-23-14539-f001:**
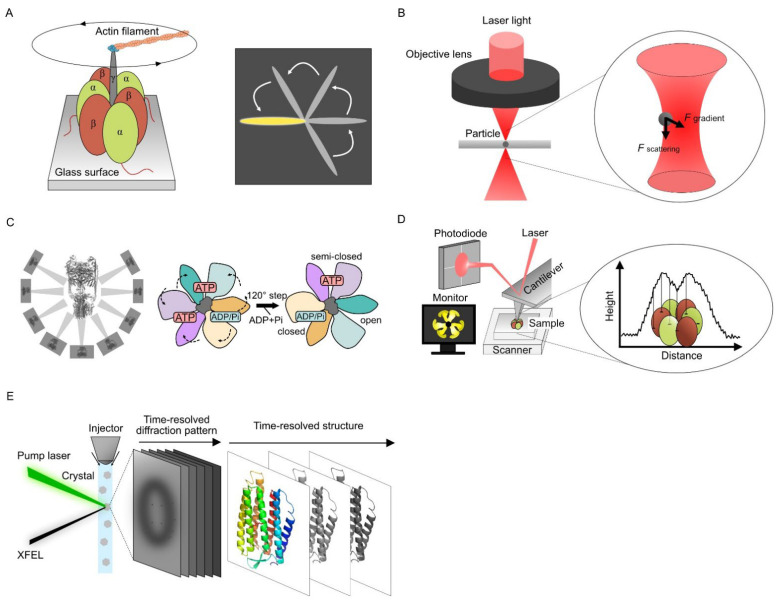
Membrane protein dynamics visualization techniques. (**A**) Visualizing single molecular dynamics using an optical microscope. The γ subunit of the F_1_ stator ring was labeled with a fluorescent actin filament. The anticlockwise rotation of the γ subunit was observed under an epifluorescence microscope [15]. (**B**) OT use a highly focused laser beam to generate force to trap or move a single molecule. This technique can be applied to understand biological mechanisms, such as conformational change, folding mechanisms, and protein-membrane interactions. (**C**) Cryo-EM and the classification technique. Two-dimensional-class averaged images of V-ATPase elucidated the rotational model of V-ATPase. ATP hydrolysis causes a 120° rotation step and conformation change in each subunit [17]. (**D**) HS-AFM scans the sample surface using the tip attached to the cantilever. The conformational changes of the protein are captured by the movement of a photodiode depending on the height of the protein. (**E**) XFEL imaging and SFX sample loading. Time-resolved X-ray diffraction and conformational changes of a protein are obtained by XFEL irradiation and photoactivation with various intervals (the image of the protein was generated from PDB:1FBK).

**Figure 2 ijms-23-14539-f002:**
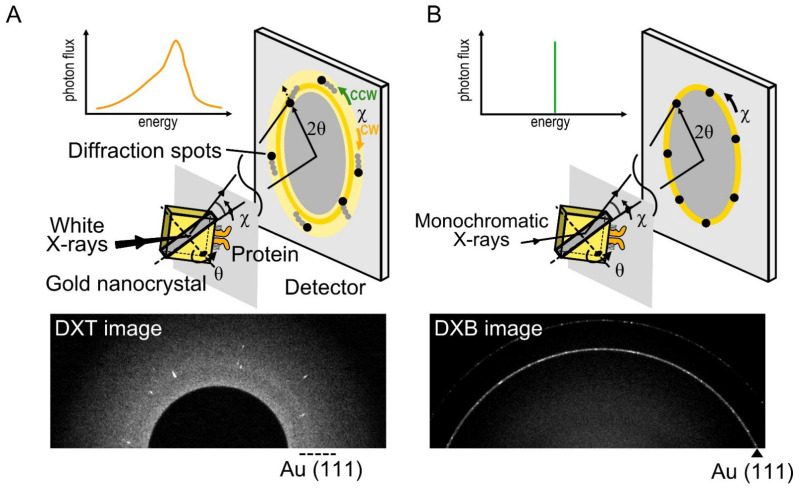
Diffracted X-ray tracking (DXT) and diffracted X-ray blinking (DXB) methods. (**A**) In DXT, the specific domain of a target protein is labeled with gold nanocrystals, with a diameter of 40 nm to 80 nm. The diffraction spots represent the tilting (*θ*) and twisting (*χ*) motions. DXT using white X-rays can analyze the trajectories of Laue spots appearing over a wide detector area. (**B**) DXB uses monochromatic X-rays, and the diffraction spots appear on the diffraction rings. The intramolecular motions are analyzed from the intensity changes of the Au (111) diffraction rings.

**Figure 3 ijms-23-14539-f003:**
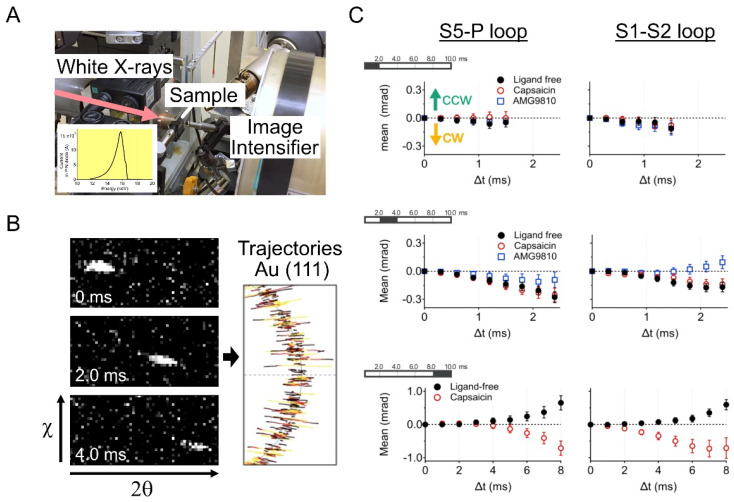
Diffracted X-ray tracking (DXT) measurement of ligand-induced rotation of TRPV1 channel. (**A**) DXT measurement system in SPring-8 BL40XU beamline. Diffraction images were recorded using an X-ray image intensifier (V5445P, Hamamatsu Photonics) and a CMOS camera (FASTCAM SA1.1, Photron, Japan). White X-rays are irradiated to the sample (inset), and the time-resolved movement diffractions are recorded. (**B**) Trajectory map of X-ray diffraction. Diffractions are obtained from Au(111) according to Bragg’s law. Trajectories move with a two-dimensional axis by tilting (θ) and twisting (χ). (**C**) Lifetime filtering enabled extracting motion components at different timescales. A CW rotational bias on TRPV1 was sustained by capsaicin (supposed to be channel opening bias) even in the longest lifetime group (8.0 ms ≦ LT < 10.0 ms). The top bars show the selected lifetime. Adapted with permission from Ref. [53]. 2020, American Chemical Society.

**Table 1 ijms-23-14539-t001:** Summary of methods for visualizing membrane protein dynamics discussed in this review.

Method	Temporal Resolution	Spatial Resolution	Labeling	Observation Area	Motion in Solution
Optical microscopy	10–100 ms	150 μm	Florescent probe	Target domain	Possible
Optical tweezers (OT)	10 μs—min	0.2 nm	Beads attached or no label	Target domain	Possible
Cryo-electron microscopy (Cryo-EM)	None (classification)	0.2 nm	No label	3D structure	Impossible (frozen)
Atomic force microscopy (AFM)	30 s—1 min	1 nm	No label	Surface structure	Impossible (needs too long time)
High-speed atomic force microscopy (HS-AFM)	ms—s	2 – 3 nm	No label	Surface structure	Possible
X-ray free electron laser (XFEL)	fs—ps	0.2 nm	No label (crystal)	3D structure	Impossible (crystal)
Diffracted X-ray tracking (DXT)	ns—ms	<0.01 nm	Gold nanocrystals (40 nm to 80 nm)	Target domain	Possible

## Data Availability

Not applicable.

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
