# Peer review of "Visualizing Intramolecular Dynamics of Membrane Proteins"

_ijms, 2022, doi:10.3390/ijms232314539_

Round 1

Reviewer 1 Report

The manuscript titled “Visualizing Intramolecular Dynamics of Membrane Proteins” by Ohkubo, T.; et al. is a work where the authors encompassed the most recent progress in the field of dynamic visualization of transmembrane proteins. Several techniques are compared like optical microscopy, cryo-electron microscopy (cryo-EM), X-ray free electron laser (XFEL), diffracted X-ray tracking (DXT) and high-speed AFM (HS-AFM) establishing well the main advantages and limitations of each one of them. Even if it is an interesting topic some improvements are required in order to increase the quality of the scientific content. The scientific paper is well written. In my opinion the present manuscript is innovative and the methodological approached used matches with the scope of International Journal of Molecular Sciences. For the above described reasons, I will recommend the publication in International Journal of Molecular Sciences once the following remarks are fixed:

--------

ABSTRACT

Abstract is clear and concise. Please, authors should introduce abbreviations in those appropriate terms like “X-ray free electron laser imaging”, “cryo-electron microscopy” or “High-speed atomic force microscopy”

--------

KEYWORDS

Please, authors should erase the numbers located at the end of each keyword according the author guidelines.

--------

INTRODUCTION

Introduction is also clear but short for a review work.

“direct detection device (DDD) cameras, which can detect electrons directly at higher frame rates than CCD cameras”. Please, define “CCD” abbreviation term by adding “charge coupled device”.

Moreover, it lacks some relevant references overall for those techniques mentioned in this section, except for HS-AFM.

“NMR applications depend on the size of molecules due to their complexity, but it can detect structure and dynamics in solution”. Reference [1] should be cited. In this work, high polarization is transferred of unpaired electrons to nuclear spins increasing, thus the sensitivity threshold providing characterization under near-physiologically conditions of membrane proteins.

“The cryo-EM single-particle analysis (SPA) has emerged as one of the most effective techniques in structural biology.” Authors should cite [2] where a graph-based algorithm was built up to generate accurate atomic models by combining EM maps with the contact predictions rendered from sequence co-evolutionary data.

“Mechanical control systems using optical tweezers now exist”. Some further reference citations should be added at the end of this sentence.

[1] Tran, N.T.; Mentink-Vigier, F.; Long, J.R. Dynamic Nuclear Polarization of Biomembrane Assemblies. Biomolecules 2020, 10, 1246. https://doi.org/10.3390/biom10091246.

[2] Bouvier, G.; Bardiaux, B.; Pellarin, R.; Rapisarda, C.; Nilges, M. Building Protein Atomic Models from Cryo-EM Density Maps and Residue Co-Evolution. Biomolecules 2022, 12, 1290. https://doi.org/10.3390/biom12091290.

Then, authors directly present HS-AFM technique: “Another approach for dynamics recording is high-speed atomic force microscopy (HS-AFM) (…)”. It may be desirable to start introducing the classical AFM and the advantages that present HS-AFM vs classical AFM. For this reason, the following statement that appears in section 2.2. Visualizing protein dynamics using high-speed atomic force microscopy: “Atomic force microscopy (AFM) was developed in 1986 to scan the surfaces of insulators of an atomic scale [22]. However, it is difficult to observe (…) takes more than 30 seconds to capture images” must be shifted to the Introduction section. Moreover, authors should add some further information that allows potential readers to better understand the capabilities of classical AFM and the advantages displayed by HS-AFM. Recent advances in the AFM field must be cited because they may be implemented to HS-AFM in the near future. In this context, sophisticated volume analyses have been developed by subtracting the background threshold providing accurate information of biomolecular association and dynamics [3]. Then, it was recently reported the hardward development to correlate AFM measurements with super-resolution fluorescence microscopy to achieve excellent lateral resolution practically in the structure illumination diffraction limit [4].

[3] Marcuello, C.; Frempong, G.A.; Balsera, M.; Medina, M.; Lostao, A. Atomic Force Microscopy to Elicit Conformational Transitions of Ferredoxin-Dependent Flavin Thioredoxin Reductases. Antioxidants 2021, 10, 1437. https://doi.org/10.3390/antiox10091437.

[4] Gómez-Varal, A.I.; Stamov, D.R.; Miranda, A.; Alves, R.; Barata-Antunes, C.; Dambournet, D.; Drubin, D.G.; Paiva, S. De Beule, P.A.A. Simultaneous-co-localized super-resolution fluorescence microscopy and atomic force microscopy: combined SIM and AFM platform for the life sciences. Sci. Rep. 2020, 10, 1122. https://doi.org/10.1038/s41598-020-57885.z.

--------

2.1. VISUALIZING SINGLE MOLECULAR DYNAMICS USING VISIBLE LIGHT

Figure 1. Authors should take care of the Figure caption format. Same comment for Figure 2 and Figure 3 (section 2.2. Diffracted X-ray tracking). Moreover, authors should indicate in the caption the PDB code of the crystal structure shown in Fig. 1D.

--------

2.2. VISUALIZING PROTEIN DYNAMICS USING HIGH-SPEED ATOMIC FORCE MICROSCOPY

Here, the authors remark the excellent lateral and temporal resolution achieved by HS-AFM and provides some tips how this technique is capable of it. Nevertheless, some further information should be furnished to improve the quality of this section.

The proper design of all HS-AFM components like scan-heads or optimized ultra-short cantilevers [5] is crucial to achieve the aforementioned goals.

[5] Valotteau, C.; Sumbul, F.; Rico, F. High-speed force spectroscopy: microsecond force measurements using ultrashort cantilevers. Biophys. Rev. 2019, 11, 689-699. https://doi.org/10.1007/s12551-019-00585-4.

The prototype and microfabrication of customized cantilevers are required to achieve fast data acquisition times (t) according to eqn. 1.

       (eqn.1)

Being, fc the cantilever resonant frequency and N the number of pixels per line.

--------

2.2. DIFRACTED X-RAY TRACKING

Table 1. Please, introduce an extra column with classical AFM in order to better visualize the differences in terms of temporal/spatial resolution, labeling, etc in comparison with HS-AFM.

--------

FUTURE PERSPECTIVES

First, authors should consider to change the current section name by “Discussion and future perspectives”.

Then, in this framework I consider relevant to state the potential of HS-AFM to address ultrafast unbinding processes of transient biomolecular complexes like streptavidin-biotin [6]. This technology can be exploited to overcome the limitations of classical AFM used for molecular recognition imaging studies [7]. The well-known above described streptavidin-biotin system can be perfectly shifted to protein membrane systems proving promising future avenues in the use of HS-AFM in this field.

[6] Rico, F.; Russek, A.; Gónzalez, L.; Grubmüller, H.; Scheuring, S. Heterogeneous and rate-dependent streptavidin-biotin unbinding revealed by high-speed force spectroscopy and atomistic simulations. Proc. Natl. Acad. Sci. U.S.A. 2019, 14, 6594-6601. https://doi.org/10.1073/pnas.1816909116.

[7] Marcuello, C.; de Miguel, R.; Lostao, A. Molecular Recognition of Proteins through Quantitative Force Maps at Single Molecule Level. Biomolecules 2022, 12, 594. https://doi.org/10.3390/biom12040594.

--------

REFERENCES

Bibliography citations are in the proper format of International Journal of Molecular Sciences. No further actions are required for this section. More references need to be added taking into account the submitted manuscript is based on a review work.

--------

OVERVIEW AND FINAL COMMENTS

The submitted work is well-designed and well-structured in the respective sections. The content is relevant and I consider it can catch the attention of potential readers and other stakeholders which may lead to gain future impact in academia. For this reason, I will recommend the present scientific manuscript for further publication in International Journal of Molecular Sciences once all the aforementioned suggestions will be properly fixed.

Reviewer 2 Report

The manuscript from Ohkubo et al presents an insightful review about observing and measuring membrane protein dynamics. The authors discuss the role of well-known techniques such as CryoEM, XFEL, HS AFMs and compare their capabilities with DXT and DXB methods for understanding the dynamics of membrane proteins. The authors highlight some key advantages of DXT technology for tracking biomolecular dynamics of membrane proteins, such as better spatiotemporal resolution, real-time dynamics measurements (as compared to molecular snapshots) and the ability to measure these events in solution. Overall, the review presents a great read to understand wide range of biophysical methods for studying protein dynamics.

I have few suggestions that might improve the outline, readability, and quality of the paper.

1.      P2, 2nd paragraph; Optical tweezers is a powerful single-molecule tool for measuring protein dynamics at the single-molecule level and can provide dynamic information in milli-micro second resolution, and often limited by the sensitivity of the force detectors (kHz vs mHz detector). The authors' reference to its limitation as the wavelength of light is not correct. It may be correct for other optical microscopy methods (for smFRET, etc), but not for optical tweezers. The authors need to explain this for better understanding.

2.      Optical tweezers (OT) has been widely used for measuring protein dynamics, as well as specifically membrane protein dynamics with high sensitivity. However, the authors presented no such cases for this review. It would be really helpful to have OT as a complementary set of tool as well as its comparison with other techniques discussed for comprehensive understanding.

https://elifesciences.org/articles/30493

https://www.nature.com/articles/ncomms10848

3.      While DXT and DXB have some intriguing advantages over conventional methods such as cryoEM and HS-AFM, it would be better to discuss some of their limitations (e.g. sample preparation, post-processing and data analysis?) to get a better perspective on different techniques and how they can be complementary to each other.

Round 2

Reviewer 1 Report

The authors have satisfactory fulfilled my suggestions. For this reason, I warmly recommend this work to further publication in International Journal of Molecular Sciences journal.